# Primary Anastomosis Versus Hartmann’s Procedure in Obstructing Colorectal Cancer: A Retrospective Cohort Study

**DOI:** 10.3390/curroncol32110636

**Published:** 2025-11-13

**Authors:** Abbas Aras

**Affiliations:** Department of General Surgery, Faculty of Medicine, Van Yuzuncu Yil University, Van 65090, Türkiye; abbasaras@yyu.edu.tr; Tel.: +90-5334158878

**Keywords:** colorectal cancer, Hartmann’s procedure, mortality, surgery, anastomosis

## Abstract

Colorectal cancer-related intestinal obstruction is a life-threatening emergency requiring urgent intervention. In this study, we compared two surgical procedures. We divided patients into two groups: 65 received Primary Resection and Anastomosis (PRA) and 45 underwent Hartmann’s Procedure (HP). We found that PRA patients experienced shorter hospital stays (8.7 vs. 11.2 days), fewer complications (33% vs. 66%), and better 5-year survival rates (33.8% vs. 22.2%). However, we observed that HP patients presented with higher baseline risk profiles, including elevated ASA scores and tumor markers, indicating they were sicker at presentation. Through multivariate analysis, we confirmed that PRA independently predicted improved survival even after adjusting for other factors. We conclude that appropriate patient selection is crucial for optimal outcomes. We recommend PRA for carefully selected patients with favorable risk profiles, while we reserve HP for high-risk scenarios. Our findings emphasize the importance of individualized surgical decision-making in emergency colorectal cancer management.

## 1. Introduction

Colorectal cancers (CRCs) are among the most common malignancies worldwide and are documented as the third most frequently diagnosed cancer type in the United States, the leading cause of cancer-related death in men, and the second leading cause in women [1,2]. The occurrence of intestinal obstruction attributable to colorectal cancer is observed in approximately 10–29% of patients presenting with an emergency condition [3]. This clinical scenario constitutes a critical emergency that necessitates prompt and decisive surgical intervention to avert grave complications such as intestinal perforations, sepsis, and mortality. The approach to surgical management and techniques is predominantly contingent upon patient-specific factors including age, performance status, tumor location, and the level of obstruction.

This exigent circumstance frequently necessitates a choice between two principal surgical methodologies in emergency decision-making contexts: Primary Resection and Anastomosis (PRA) and Hartmann’s Procedure (HP). PRA encompasses resection of the obstructed bowel segment followed by anastomosis, facilitating immediate restoration of intestinal continuity. Conversely, HP entails resection accompanied by establishment of a proximal colostomy; this approach is typically reserved for patients at elevated operative risk or in situations where anastomosis is deemed unsafe. The rates of postoperative complications associated with primary surgical interventions are reported to be 7.3% for mortality and between 21.5% and 54.9% for overall morbidity [4,5]. Despite the growing acceptance of PRA in recent years due to their capacity to mitigate complications associated with end-colostomy, controversies and inconclusive findings persist. A significant concern remains the risk of anastomotic leakage, which may pose a life-threatening condition. To address this risk, a diverting ileostomy is frequently conducted to safeguard the anastomosis.

This retrospective investigation sought to elucidate this problem by comparing the clinical characteristics, perioperative conditions and oncological outcomes of patients undergoing to PRA versus HP in an emergency context. Through the assessment of early postoperative complications, morbidity and mortality rates, and survival outcomes extending to five years, we aspire to furnish critical insights that can enhance surgical decision-making in the management of obstructive CRC.

## 2. Methods

This investigation was conducted as a retrospective cohort study within a tertiary referral university hospital. The cohort comprised patients diagnosed with obstructed colorectal cancer who underwent emergency surgical intervention between May 2014 and August 2019. Exclusion criteria included patients who developed intestinal perforation with generalized peritonitis, those with prior surgical history related to colorectal cancer, and individuals who experienced colorectal obstruction attributable to malignancies of other origins or benign etiologies. Patients presenting with frank peritonitis were excluded to avoid confounding from severe septic conditions that would have mandated HP regardless of other factors. The institutional ethics review board granted approval for this study, with approval number 03, on 25 September 2024. All methodologies employed throughout the study conformed to the ethical principles outlined in the Declaration of Helsinki. Participants, or their designated relatives, were contacted through phone calls to secure informed consent for study participation. Due to the retrospective nature of this study involving data collected 5–10 years previously, and considering the challenges of obtaining written consent from geographically dispersed patients (many living in remote regions), the ethics committee approved verbal consent obtained via telephone as documented in the study protocol. Verbal consent was systematically recorded and documented in study files, with patients (or legal representatives for deceased patients) informed about study aims, data usage, and right to withdraw. This approach aligned with institutional guidelines for retrospective minimal-risk research and was approved by the ethics committee (approval number 25.09.2024/03).

To minimize selection bias, all consecutive patients who met inclusion criteria during the study period were enrolled, and their data were verified against operative and pathology logs. Data abstraction was performed by two independent investigators using a standardized form to ensure accuracy and completeness. Data were systematically extracted from hospital medical records, including demographic information (age, sex, body mass index), surgical intervention type, pathological findings, TNM tumor staging, American Society of Anesthesiologists (ASA) classification, early postoperative complications, hospital readmissions, need for subsequent surgeries, morbidity and mortality rates (30-day, 1-year, and 5-year survival), and renal function assessed by serum creatinine and urea. Early complications were identified as wound infections, anastomotic leakage, sepsis, and pulmonary embolism. Furthermore, long-term oncological outcomes, including local recurrence, distant metastasis, and disease-free survival, were systematically evaluated. Missing data were rare (<5% for biochemical parameters) and were excluded from analysis on a case-by-case basis without imputation.

## 3. Surgical Procedures

Selection of PRA or HP was guided by established clinical principles and intraoperative assessment. PRA was preferentially performed in hemodynamically stable patients with adequate physiological reserve, absence of gross peritoneal contamination, satisfactory bowel perfusion and quality for safe anastomosis, and ASA I-II classification when feasible. HP was reserved for specific high-risk scenarios including hemodynamic instability requiring damage control surgery, presence of significant peritonitis (note: patients with frank perforation and generalized peritonitis were excluded from this study), unresectable tumor at the time of surgery, distal obstruction preventing safe anastomosis, or suspected colonic ischemia precluding reliable anastomotic healing. The final surgical decision was individualized based on the operating surgeon’s intraoperative assessment of patient physiology, tumor characteristics, and technical feasibility.

Patients were divided into two distinct cohorts based on the surgical approach undertaken: Group A (Primary Resection and Anastomosis—PRA) and Group B (Hartmann’s Procedure—HP). In the PRA cohort, resection of the obstructed segment was followed by anastomosis to re-establish bowel continuity. Conversely, in the HP cohort, resection was followed by a proximal colostomy without subsequent anastomosis. For select patients within the PRA cohort, a temporary ileostomy was established to safeguard the anastomosis site. All surgical interventions were conducted via open laparotomy, with all surgeons affiliated with their individual institutions being well-equipped and possessing substantial experience in colorectal malignancy surgeries.

The diagnostic modalities employed preoperatively included Computed Tomography (CT), colonoscopy, and/or abdominal Magnetic Resonance Imaging (MRI). As this was a retrospective study, sample size was determined by the number of patients meeting inclusion criteria during the study period (May 2014 to August 2019).

## 4. Statistical Analysis

Descriptive statistics for the examined variables (characteristics) were presented as mean, standard deviation, and minimum and maximum values. A chi-square test was conducted to assess the association between the two categorical variables. Odds ratios were used for further statistical analysis. Survival analysis was conducted to determine the median survival duration for the respective groups. The Kaplan–Meier method was used to estimate the survival function from the lifetime data. Multivariate Cox proportional hazard regression was performed to assess independent predictors of survival. The model included surgical approach (PRA vs. HP), ASA classification, CEA levels (using cutoff 5 ng/mL), CA 19-9 levels (using cutoff 37 U/mL), tumor localization (right vs. left), and age as covariates. Hazard ratios (HR) with 95% confidence intervals (CI) were calculated. The model fit was assessed using −2 Log Likelihood and overall model significance was tested using omnibus chi-square tests. Furthermore, statistical significance was set at 5%, and the SPSS software (version 13) was utilized for all statistical computations.

The patient selection process, inclusion and exclusion criteria, and final cohort distribution are illustrated in Figure 1. The flow diagram follows STROBE recommendations for observational studies to ensure transparency of participant selection and data analysis steps (Appendix A).

## 5. Results

### 5.1. Demographic Characteristics and Perioperative Outcomes

Between 1 May 2014 and 30 August 2019, this study included 110 patients who underwent urgent surgical intervention for obstructive colorectal neoplasia, irrespective of its anatomical location. Among these patients, 65 individuals underwent Primary Resection and Anastomosis (PRA, Group A), while 45 patients received Hartmann’s Procedure (HP, Group B).

The mean age of the entire cohort was 60.3 ± 15.2 years. The distribution of ASA classification indicated a statistically significant predominance of high-risk patients (ASA III-IV) within the HP group compared with the PRA group (66% vs. 36%, *p* = 0.040) (Table 1).

The laboratory parameters presented in Table 1 and Table 2 demonstrated significant biochemical differences between the PRA and HP groups. Tumor biomarkers, specifically carcinoembryonic antigen (CEA) and CA 19-9, were markedly elevated in the HP cohort (*p* = 0.009 and *p* = 0.004, respectively), suggesting that patients in the HP group may have presented with more advanced or aggressive malignancies. Albumin concentrations were higher in the PRA cohort (3.40 ± 0.76 g/dL) than in the HP group (3.01 ± 0.87 g/dL), although this difference did not reach statistical significance (*p* = 0.105). Renal function parameters suggested higher acute physiological stress in the HP group, with elevated urea levels (43.04 ± 22.37 vs. 33.51 ± 16.67 mg/dL, *p* = 0.024), though creatinine levels were similar between groups (0.92 ± 0.37 vs. 0.81 ± 0.21 mg/dL, *p* = 0.317). Mean operative time was significantly longer for PRA compared to HP (122.37 ± 37.10 vs. 99.67 ± 45.56 min, Cohen’s d = 0.56, *p* = 0.003).

Pathological T-stage distribution showed that T3 disease was most common in both surgical groups (PRA: 69.2%, HP: 64.4%). T4 disease, representing locally advanced tumors, was present in 16 PRA patients (24.6%) and 16 HP patients (35.6%), while T2 disease was documented in 4 PRA patients (6.2%) with no T2 cases in the HP group. Regarding nodal status, lymph node involvement (N1-N3) was observed in 32 PRA patients (49.2%) compared to 29 HP patients (64.4%). Distant metastasis (M1) was present in four PRA patients (6.2%) and three HP patients (6.7%). Overall, HP patients demonstrated a trend toward more advanced local disease (higher T4 and N+ rates), though both groups included patients across the spectrum of disease stages. Adjuvant chemotherapy was administered to 36 PRA patients (55.4%) compared to 17 HP patients (37.8%, *p* = 0.069).

Patients in the PRA cohort experienced significantly shorter hospital stay (8.7 ± 4.1 vs. 11.2 ± 5.2 days, Cohen’s d = 0.55, *p* = 0.02) and had lower complication rates (33% vs. 66%, OR = 0.26 *p* = 0.003). Patients in the PRA group had significantly better survival outcomes, with a mean survival duration of 1969 ± 1113 days compared with 626 ± 759 days for the HP cohort (*p* = 0.001).

### 5.2. Comparison of PRA Cases with and Without Diverting Ostomy

Within the PRA cohort, 4 patients underwent diverting ostomy, while 61 did not (Table 2). Exploratory analysis of PRA subgroups revealed several trends, though the very small ostomy sample size (*n* = 4) precludes definitive conclusions.

### 5.3. Outcomes Based on Tumor Localization

The management of patients with right-sided tumors was predominantly conducted through PRA (30% in the PRA cohort versus 6% in the HP cohort, *p* = 0.002). In contrast, patients with left-sided tumors more frequently underwent HP treatment procedures (70% PRA versus 93% HP, *p* = 0.001) (Table 1).

### 5.4. Comparison of Subgroup Outcomes

The long-term oncologic outcomes were further scrutinized among the delineated subgroups, categorized as Group A1 (PRA without ostomy), Group A2 (PRA with ostomy), Group B1 (HP with resection), and Group B2 (HP without resection). These findings are encapsulated in Table 3. The primary discrepancies noted in the analysis revealed that patients within Group A2 exhibited a diminished mean duration of hospitalization (10.2 ± 3.1 days) in contrast to Group A1 (8.89 ± 4.18 days) and Group B1 (16.6 ± 9.6 days). Exploratory subgroup analysis examined outcomes across four groups, though unequal sample sizes—particularly the small A2 group (*n* = 4)—limit definitive interpretation.

### 5.5. Long-Term Oncologic Outcomes

The synthesis of long-term oncologic outcomes between the PRA and HP groups is encapsulated in Table 4. Recurrence rates were minimal in both groups, revealing no statistically significant difference (3.1% in the PRA group compared to 6.7% in the HP group, *p* = 0.37). In a similar vein, metastasis rates were found to be comparable (35% in PRA versus 37% in HP, *p* = 0.96).

Nonetheless, marked differences were discerned in survival outcomes. The 1-year survival rate was significantly elevated in the PRA group in comparison to the HP group (84.6% versus 66.6%, *p* = 0.001). The PRA cohort exhibited a significantly improved 5-year survival rate relative to the HP cohort (33.8% versus 22.2%; *p* = 0.003, HR = 0.72, 95% CI: 0.55–0.90). However, the 30-day survival rate did not reveal any significant differences between the groups (96.9% in PRA versus 91.1% in HP, *p* = 0.46). Individuals in the PRA group were found to have significantly improved survival outcomes, with a mean survival duration of 1969 ± 1113 days in contrast to 626 ± 759 days for the HP cohort (Cohen’s d = 1.38, very large effect, *p* = 0.001).

Multivariate Cox regression analysis was performed to identify independent predictors of survival while adjusting for baseline imbalances (Table 5). The overall model was statistically significant (chi-square = 15.967, df = 6, *p* = 0.014), indicating that the combined variables significantly predicted survival. In the adjusted model, surgical approach remained a significant independent predictor of survival. PRA was associated with improved survival compared to HP (HR = 0.332, 95% CI: 0.157–0.702, *p* = 0.004), indicating PRA patients had approximately 67% lower risk of death after adjusting for confounders. However, none of the other covariates—including ASA score (HR = 0.987, *p* = 0.953), CEA > 5 ng/mL (HR = 1.000, *p* = 0.523), CA 19-9 > 37 U/mL (HR = 1.000, *p* = 0.976), tumor localization (HR = 1.445, *p* = 0.382), or age (HR = 1.016, *p* = 0.174)—reached statistical significance as independent predictors.

## 6. Discussion

This study compared perioperative and long-term outcomes of PRA versus HP in emergency surgery for obstructing colorectal cancer. Our principal findings demonstrate that PRA was associated with shorter hospital stays (8.7 vs. 11.2 days, *p* = 0.02), fewer complications (33% vs. 66%, *p* = 0.003), and superior survival outcomes (5-year survival: 33.8% vs. 22.2%, *p* = 0.003) compared with HP (Figure 2). However, these differences must be interpreted within the context of significant baseline differences between groups, with HP patients presenting with higher ASA scores (66% vs. 36% ASA III–IV, *p* = 0.040), elevated tumor markers, and greater physiological derangement at presentation.

Recent comprehensive reviews emphasize that surgical decision-making in obstructive colorectal cancer must be individualized rather than formulaic, with technical decisions regarding vascular ligation level, extent of lymphadenectomy, and surgical approach tailored to individual patient anatomy, tumor characteristics, and physiological status rather than rigid adherence to a single protocol [6]. This patient-centered, multidimensional approach—integrating surgeon expertise, patient factors, surgical technique, and available technology—aligns with our findings that appropriate surgical selection, rather than a universally superior technique, determines optimal outcomes in emergency colorectal surgery. By excluding patients with frank peritonitis, our study focused on cases where either PRA or HP was theoretically feasible, allowing meaningful evaluation of surgical choice when both approaches remain viable options.

In carefully selected patients with favorable risk profiles, PRA offers significant advantages by preserving bowel continuity and avoiding permanent stoma formation. Our findings corroborate existing literature demonstrating reduced morbidity and shorter hospital stays with PRA when patients are selected based on hemodynamic stability and tumor characteristics [7,8]. However, anastomotic leakage is a serious complication, with rates of 8–20% reported in emergency settings and associated mortality of 10–25% [9,10,11]. Newer techniques for anastomotic assessment, including intraoperative colonoscopy and indocyanine green fluorescence imaging [12,13], can identify high-risk anastomoses and inform decisions about protective diversion. Contemporary technical refinements, including fully intracorporeal approaches in robotic surgery [14] and comprehensive assessment protocols [15], demonstrate ongoing evolution that may further optimize outcomes in both elective and emergency settings.

Hartmann’s Procedure is essential for high-risk patients when survival takes priority over avoiding a stoma. HP is the appropriate choice for patients with hemodynamic instability, severe peritoneal contamination, poor bowel viability, or major comorbidities that would make anastomotic failure particularly dangerous. Although our HP cohort included mostly ASA III-IV patients with elevated tumor markers and greater physiological stress, the 91.1% 30-day survival rate confirms that HP is the right choice when patient safety requires damage-control surgery. While HP is associated with prolonged recovery, higher complication rates, and reduced quality of life related to permanent stoma formation [9,11,16,17], these drawbacks are acceptable trade-offs when anastomosis poses unacceptable risk. Reports indicating that 40–60% of patients never undergo colostomy reversal [16,17] underscore the importance of appropriate patient selection to maximize the feasibility of PRA when safe.

PRA demonstrated superior survival at both one year (84.6% vs. 66.6%, *p* = 0.001) and five years (33.8% vs. 22.2%, *p* = 0.003), with a very large effect size (Cohen’s d = 1.38). However, these survival differences must be interpreted cautiously given the similar recurrence (3.1% vs. 6.7%, *p* = 0.37) and metastasis rates (35% vs. 37%, *p* = 0.96) between groups. This paradox—improved survival without differences in disease recurrence—suggests that survival advantages primarily reflect perioperative mortality differences and patient selection rather than superior oncological disease control [18,19,20,21]. PRA patients’ better baseline physiological reserve enabled survival of complications and receipt of adjuvant therapy (55.4% vs. 37.8%, *p* = 0.069), whereas HP patients with similar tumor biology succumbed to perioperative complications and comorbidities. TNM staging differences, with HP patients showing more advanced T4 disease (35.6% vs. 24.6%) and higher nodal involvement (64.4% vs. 49.2%), indicate that higher-risk patients were appropriately selected for damage-control surgery.

Multivariate Cox regression demonstrated that surgical approach remained an independent predictor of survival (adjusted HR = 0.332, 95% CI: 0.157–0.702, *p* = 0.004) after adjusting for ASA classification, tumor markers, tumor location, and age. However, several important limitations should be noted. Traditional risk factors (ASA, CEA, CA 19-9) did not reach statistical significance in the multivariate model, likely reflecting our small sample size and retrospective data limitations. Additionally, we could not adjust for critical acute parameters such as hemodynamic stability, SIRS criteria, and bowel perfusion that influenced surgical selection. The substantial baseline differences between groups—particularly in ASA scores, tumor markers, and renal function—indicate that selection bias persists despite statistical adjustment. Therefore, the observed survival advantage likely reflects appropriate patient selection and surgical matching rather than universal PRA superiority. Our findings show that the best outcomes come from matching the surgical technique to each patient’s risk profile, not from applying either approach. Our study contributes real-world data on emergency colorectal surgery outcomes with five-year follow-up. Much of the existing literature focuses on elective procedures or lacks long-term oncological data [22,23,24,25,26,27]. The exclusion of peritonitis cases represents a methodological strength rather than limitation, as it allowed comparison in patients in whom both surgical approaches remained viable options. In clinical practice, peritonitis mandates HP regardless of other factors; excluding these cases enabled meaningful evaluation of surgical decision-making based on patient risk stratification. Our multivariate analysis attempts to account for baseline imbalances, though residual confounding remains likely. The subgroup analyses of diverting ostomy (*n* = 4) and HP with or without resection are exploratory given the small sample sizes, but they suggest directions for future research on protective diversion strategies and surgical extent in high-risk patients.

This study has several important limitations. The retrospective single-center design limits generalizability and introduces potential selection bias, particularly in surgical approach allocation. We lacked detailed acute physiological parameters at the time of surgery (hemodynamic status, SIRS criteria, real-time laboratory markers) that undoubtedly influenced surgical decisions and outcomes. The significant baseline differences between groups—despite statistical adjustment—indicate residual confounding that prevents definitive causal inference about surgical technique superiority. Small sample sizes in subgroup analyses (particularly *n* = 4 for PRA with ostomy) preclude robust statistical conclusions. We did not evaluate cost-effectiveness or detailed quality-of-life outcomes, which are important patient-centered measures. The absence of data on specific anastomotic techniques, diversion strategies, and surgeon-specific factors limits our ability to identify technical factors influencing outcomes.

In conclusion, PRA was associated with reduced hospital stays, lower complication rates, and enhanced survival rates, highlighting its potential advantages in preventing permanent stoma formation when appropriate patient selection criteria are met. In contrast, HP continues to represent a viable alternative for high-risk patients or scenarios in which anastomosis is considered unsafe. These findings underscore the critical nature of individualized surgical decision-making in emergency colorectal cancer conditions.

Future research should prioritize prospective multicenter registries with standardized patient selection criteria and systematic documentation of acute physiological parameters to minimize selection bias. Investigation of modern diversion strategies, including temporary stenting as bridge to surgery, may refine emergency management algorithms. Patient-centered outcomes including quality of life assessments, stoma-related morbidity, and cost-effectiveness analyses are essential to comprehensively evaluate these surgical approaches. Ultimately, development of validated risk stratification tools incorporating tumor biology, physiological status, and surgical factors would enable evidence-based, individualized surgical decision-making in this challenging clinical scenario.

## Figures and Tables

**Figure 1 curroncol-32-00636-f001:**
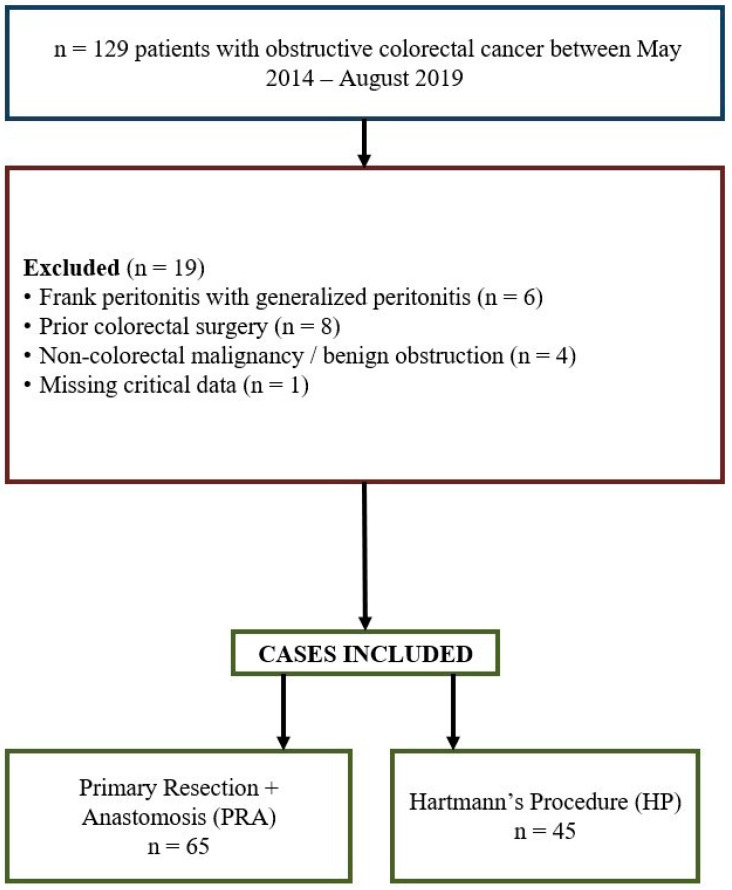
Flow diagram showing patient selection process and allocation to surgical procedures. Of 129 patients presenting with obstructing colorectal cancer, 19 were excluded due to predefined criteria, leaving 110 eligible cases. These were divided into the Primary Resection and Anastomosis (*n* = 65) and Hartmann’s Procedure (*n* = 45) groups for analysis.

**Figure 2 curroncol-32-00636-f002:**
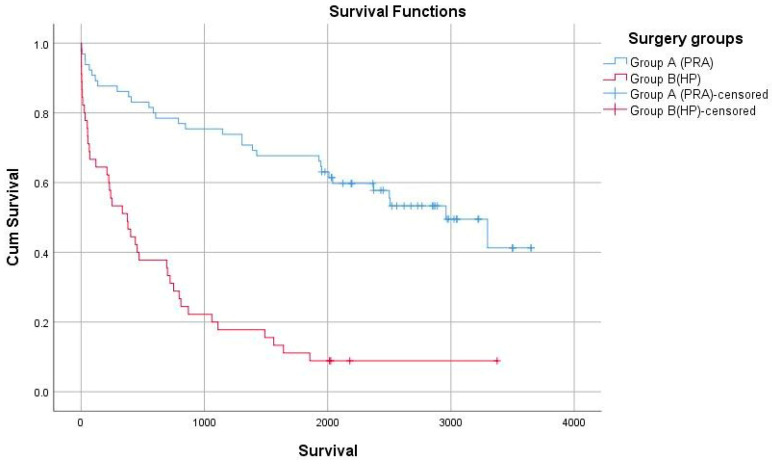
The Kaplan–Meier survival analysis, accentuating a significant divergence in survival rates between the two surgical cohorts. Group A (Primary Resection and Anastomosis—PRA) demonstrates a markedly superior long-term survival in comparison to Group B (Hartmann’s Procedure—HP) (*p*: 0.002).

**Table 1 curroncol-32-00636-t001:** Demographic characteristics, clinical data and perioperative outcomes between groups.

Variables	Group A (PRA) *n* = 65	Group B (HP) *n* = 45	*p*	Effect Size
	*Mean ± SD*	*Mean ± SD*		
Age [year]	57.06 ± 16.6	65.09 ± 16.5	0.560	0.49
Survival [day]	1969 ± 1113	626 ± 759	**0.001**	1.38
Hospitalization days	8.7 ± 4.1	11.2 ± 15.2	**0.020**	0.55
** *Laboratory parameters* **	*Mean ± SD*	*Mean ± SD*	*p*	
WBC [×10^3^/μL]	9.79 ± 3.51	9.31 ± 4.77	0.546	0.12
Hb [g/dL]	12.24 ± 2.29	12.44 ± 2.37	0.660	0.09
Alb	3.40 ± 0.765	3.01 ± 0.871	0.105	0.48
CA 19-9	25.33 ± 43.06	216.11 ± 440.721	**0.004**	0.67
CEA	9.06 ± 17.84	204.07 ± 507.27	**0.009**	0.56
CRP [mg/L]	58.29 ± 51.48	64.87 ± 73.78	0.619	0.11
Creatinine	0.81 ± 0.21	0.92 ± 0.37	0.317	0.38
Urea	33.51 ± 16.67	43.04 ± 22.37	**0.024**	0.49
** *ASA Score* **	** *(n%)* **	** *(n%)* **		
I-II	42 (64%)	15 (33%)	**0.040**	3.65
III-IV	23 (36%)	30 (66%)		
** *Tumor localization* **	** *(n%)* **	** *(n%)* **		
Right sided	20 (30%)	3 (6%)	**0.002**	6.22
Left sided	45 (70%)	42 (93%)		
** *Number of metastasis* **	** *(n%)* **	** *(n%)* **		
1 organ	19 (29.2%)	14 (31.1%)	0.967	0.90
2 or more organs	4 (6.2%)	3 (6.7%)		
Need for repeat surgery	16 (24.6%)	20 (44.4%)	**0.029**	0.41
Complications	22 (33%)	30 (66%)	**0.003**	0.26
Gender [M/F] *n*/%	[31/34, (48%/52%)	25/20, (56%/44%)	0.650	0.73
Surgery Time	122.37 ± 37.10	99.67 ± 45.56	**0.003**	0.56
Adjuvant therapy				
Present	36 (55.4%)	17 (37.8%)	0.069	2.04
Absent	29 (44.6%)	28 (62.2%)		

PRA: Primary Resection and Anastomosis, HP: Hartmann’s Procedure, ASA: American Society of Anesthesiology, CEA: carcinoembryonic antigen, CRP: C-reactive protein. *p* < 0.05 indicates statistical significance.

**Table 2 curroncol-32-00636-t002:** Comparison of peri-operative outcomes of PRA cases with and without diverting ostomy.

Outcomes Value	PRA Case with Ostomy *n* = 4	PRA Case Without Ostomy *n* = 61
	Mean ± SD	Mean ± SD
Age, years	62 ± 16.9	56.7 ± 16.2
Hemoglobin	14.77 ± 1.75	12.07 ± 2.24
Albumin	2.60 ± 1.69	3.47 ± 0.67
CRP	84.57 ± 68.84	55.95 ± 49.99
Hospitalization days	10.2 ± 3.1	7.1 ± 2.1
** *ASA Score* **	*n* (%)	*n* (%)
I-II	4 (100%)	38 (62%)
III-IV	0	23 (38%)
** *Tumor localization* **		
Right-sided	0	20 (32%)
Left-sided	4 (100%)	41 (67%)
** *Complications* **		
Surgical site infection	0	5 (8.2%)
Multiorgan failure	0	0
Anastomosis leakage	0	5 (8.2%)
Re-operation	3 (75%)	13 (21.3%)
Mortality	0	2 (3.3%)

PRA: Primary Resection and Anastomosis. Due to small sample size in the ostomy group (*n* = 4), statistical comparisons should be interpreted with extreme caution and are presented for descriptive purposes only. These findings are hypothesis-generating and require validation in larger cohorts.

**Table 3 curroncol-32-00636-t003:** Comparison of subgroup outcomes: exploratory analysis.

Variable	Group A1	Group A2	Group B1	Group B2
Hospitalization day, [Mean ± SD]	8.89 ± 4.18	7 ± 2.5	16.6 ± 9.6	6.04 ± 6.35
Recurrence [*n* (%)]	2 (3.3%)	0	1 (4.5%)	2 (8.7%)
Metastasis rates [*n* (%)]	23 (37.7%)	0	6 (27.3%)	11 (47.8%)
Complications [*n* (%)]	19 (31.1%)	3 (75%)	19 (86%)	11 (47.8%)
Need for repeat surgery [*n* (%)]	13 (21.3%)	3 (75%)	13 (59.1%)	7 (30.4%)
Tumor localization [Right sided, *n* (%)]	20 (32.8%)	0	2 (9.1%)	1 (4.3%)
Tumor localization [Left sided, *n* (%)]	41 (67.2%)	4 (100%)	20 (90.9%)	22 (95.7%)
Co-morbidities [*n* (%)]	14 (22.9%)	0	4 (18%)	7 (30.4%)
Surgical site infection [*n* (%)]	5 (8.2%)	0	3 (13.6%)	2 (8.7%)
1-Year survival rate (%)	80.32%	100%	77.2%	65.2%
5-Year survival rate (%)	40.9%	25%	36.3%	26%

Group A2 (PRA with ostomy, *n* = 4) involves very small sample size. Statistical tests involving this group should be interpreted cautiously.

**Table 4 curroncol-32-00636-t004:** Comparison of long-term oncologic outcomes between PRA and HP groups.

Variables	Group A (PRA) *n* = 65	Group B (HP) *n* = 45	*p*
	*n* (%)	*n* (%)	
Recurrence	2 (3.1%)	3 (6.7%)	0.370
Metastasis rates	23 (35%)	17 (37%)	0.960
** *Survival* **	%	%	*p*
30-day survival rate	96.9	91.1	0.460
1-Year survival rate	84.6	66.6	**0.001**
5-Year survival rate	33.8	22.2	**0.003**

PRA: Primary Resection and Anastomosis, HP: Hartmann’s Procedure. *p* < 0.05 indicates statistical significance.

**Table 5 curroncol-32-00636-t005:** Multivariate Cox proportional hazard regression model for survival.

Variable	B	SE	*p* Value	HR	95% CI
Surgery groups (PRA vs. HP)	−1.102	0.382	0.004	0.332	0.157–0.702
ASA score	−0.013	0.225	0.953	0.987	0.635–1.533
CEA	0.001	0.001	0.523	1.000	0.999–1.002
CA 19-9	0.001	0.001	0.976	1.000	0.998–1.002
Tumor localization	0.368	0.421	0.382	1.445	0.633–3.298
Age	0.016	0.012	0.174	1.016	0.993–1.039

B: regression coefficient; SE: standard error; HR: hazard ratio; CI: confidence interval. Reference categories: HP for surgery groups, left-sided for tumor localization. Overall model: χ^2^ = 15.967, df = 6, *p* = 0.014; −2 Log Likelihood = 286.919.

## Data Availability

The data of the study can be provided by the corresponding author upon request.

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
