# Peer review of "Primary Anastomosis Versus Hartmann’s Procedure in Obstructing Colorectal Cancer: A Retrospective Cohort Study"

_curroncol, 2025, doi:10.3390/curroncol32110636_

Round 1
Reviewer 1 Report
Comments and Suggestions for Authors
Dear colleague,
Congratulations on the paper. It has been a huge effort to collect and thoroughly analyze the data. I understand that you have collected much more data than presented in the publication, and I would like some clarifications:
Are some criteria for performing one type of surgery or another?
Patient characteristics have been analyzed in terms of their baseline status according to ASA, but I believe that the patients' acute condition at the time of surgery should also be analyzed: hemodynamic stability, SIRS, renal function, leukocytosis. This may influence the indication for the type of surgery and also the short-term morbidity and mortality outcomes. I would also rule out a possible population bias to rule out comparing very different patient groups.
And I would also rule out the presence or absence of associated peritonitis.
Surgical times are also not analyzed.
Was it possible to collect the staging of the resected lesions? Could the patients have received adjuvant treatment or not? This could affect long-term results.
Awaiting clarification.
Congratulations again.
Comments on the Quality of English Language
I think the quality of English language is correct.
Reviewer 2 Report
Comments and Suggestions for Authors
This retrospective single-center study compares outcomes between Primary Resection and Anastomosis (PRA) and Hartmann’s Procedure (HP) in the emergency management of obstructing colorectal cancer. A total of 110 patients treated between 2014–2019 were analyzed: 65 underwent PRA and 45 HP. Results indicated that PRA was associated with significantly shorter hospital stay, fewer complications, and improved long-term survival (1-year and 5-year survival rates) compared to HP. Subgroup analyses also examined the role of diverting ostomy within PRA and resection status in HP. The authors conclude that PRA, when feasible, offers superior perioperative and oncological outcomes and reduces the burden of permanent stoma formation. While the study addresses a clinically relevant and timely question, the manuscript requires major revision before being considered for publication. The topic—emergency surgical strategies in obstructing colorectal cancer—is of high interest, but several methodological, statistical, and interpretive issues weaken the current version. Furthermore, contextualization within the broader literature is incomplete, and certain key methodological aspects are underreported. Below I outline specific areas requiring substantial improvement.
-
The work is retrospective and single-center, yet the introduction and discussion sometimes present findings with a causal tone, as if deriving from a randomized design. The limitations should be highlighted earlier and more explicitly, not relegated to the final paragraphs.
-
The HP group includes significantly more ASA III–IV patients and elevated tumor markers (CEA, CA 19-9). This imbalance undermines comparability between groups. The authors should discuss whether these differences reflect surgical selection bias, and ideally perform multivariate or propensity-matched analyses to adjust for confounders. Currently, comparisons may simply reflect baseline differences rather than true surgical superiority.
-
The manuscript mentions a “t-test for sample size determination” (Methods) but does not clarify whether a formal power calculation was performed before data collection (unlikely in a retrospective study). This section requires rewriting.
-
Many p-values are reported without effect sizes or confidence intervals. For survival analysis, hazard ratios and confidence intervals should be provided consistently.
- The subgroup analyses (e.g., PRA with ostomy, HP with/without resection) involve very small sample sizes (n=4 for PRA with ostomy), which makes percentages misleading and statistical testing questionable. These should be reported descriptively, with clear caution about overinterpretation.
-
The Kaplan–Meier curves are mentioned but not adequately discussed. The dramatic differences in mean survival (1969 vs. 626 days) likely reflect selection bias. The authors should emphasize that survival advantages may not solely be attributable to the surgical approach but also to underlying disease burden and comorbidity.
-
Recurrence and metastasis rates are reported as “not significantly different,” yet the authors claim oncological superiority for PRA. This is inconsistent. A more cautious interpretation is warranted: while PRA patients lived longer, recurrence patterns were similar, suggesting survival differences may reflect patient selection and perioperative risk rather than tumor biology.
-
The discussion reads as overly favorable towards PRA. Greater acknowledgment is needed of the risks of anastomotic leakage and the rationale for HP in high-risk patients. The literature review should be expanded to include contemporary guidelines and systematic reviews on emergency colorectal obstruction. The recent review discusses vascular ligation level, decision-making in colorectal surgery, and balancing oncological versus functional outcomes. Linking this to the current study would strengthen the argument that vascular and technical decisions (e.g., PRA vs. HP, ligation level) must be individualized rather than formulaic (doi: 10.3390/cancers16010072).
- Tables require clearer formatting, especially subgroup analyses. Denominators should always be explicit.
-
Some variables (e.g., CA 19-9, CEA) show striking differences between groups; their clinical interpretation is not adequately discussed.
-
Terminology should be standardized (e.g., “exitus” should be replaced by “death” or “mortality”).
-
Verbal informed consent obtained via phone calls, years after the operation, should be justified in more detail, as this could raise ethical concerns in an international readership.
-
The manuscript contains grammatical inconsistencies and sometimes cumbersome phrasing. A thorough language revision by a native speaker or professional editing service is strongly recommended to improve readability.
-
The conclusion should not only restate findings but also suggest future research directions: prospective multicenter registries, use of modern diversion strategies, quality of life assessments, and cost-effectiveness analyses. These aspects would make the contribution more compelling.
Round 2
Reviewer 2 Report
Comments and Suggestions for Authors
The Author made a great revision addressing all concerns raised by referees. Now the paper has been notably improved and it deserves to be published in its current form.
Author Response
Thank you for your comments.